# Comparing Genetic and Physical Anthropological Analyses for the Biological Profile of Unidentified and Identified Bodies in Milan

**DOI:** 10.3390/genes14051064

**Published:** 2023-05-11

**Authors:** Elena Pilli, Andrea Palamenghi, Alberto Marino, Nicola Staiti, Eugenio Alladio, Stefania Morelli, Anna Cherubini, Debora Mazzarelli, Giulia Caccia, Daniele Gibelli, Cristina Cattaneo

**Affiliations:** 1Laboratorio di Antropologia Molecolare Forense, Dipartimento di Biologia, Università degli Studi di Firenze, Via del Proconsolo 12, 50122 Florence, Italy; 2LABANOF—Laboratorio di Antropologia e Odontologia Forense, Dipartimento di Scienze Biomediche per la Salute, Università degli Studi di Milano, Via L. Mangiagalli 37, 20133 Milan, Italy; 3LAFAS—Laboratorio di Anatomia Funzionale dell’Apparato Stomatognatico, Dipartimento di Scienze Biomediche per la Salute, Università degli Studi di Milano, Via L. Mangiagalli 31, 20133 Milan, Italy; 4Reparto Carabinieri Investigazioni Scientifiche di Parma, Sezione Biologia, 43121 Parma, Italy; 5Department of Chemistry, University of Turin, 10124 Turin, Italy

**Keywords:** forensic anthropology, skeletal remains, sex, ancestry, autosomal and Y STRs, mitochondrial genome

## Abstract

When studying unknown human remains, the estimation of skeletal sex and ancestry is paramount to create the victim’s biological profile and attempt identification. In this paper, a multidisciplinary approach to infer the sex and biogeographical ancestry of different skeletons, using physical methods and routine forensic markers, is explored. Forensic investigators, thus, encounter two main issues: (1) the use of markers such as STRs that are not the best choice in terms of inferring biogeographical ancestry but are routine forensic markers to identify a person, and (2) the concordance of the physical and molecular results. In addition, a comparison of physical/molecular and then antemortem data (of a subset of individuals that are identified during our research) was evaluated. Antemortem data was particularly beneficial to evaluate the accuracy rates of the biological profiles produced by anthropologists and classification rates obtained by molecular experts using autosomal genetic profiles and multivariate statistical approaches. Our results highlight that physical and molecular analyses are in perfect agreement for sex estimation, but some discrepancies in ancestry estimation were observed in 5 out of 24 cases.

## 1. Introduction

When unidentified human remains are recovered, forensic investigators are tasked with creating a biological profile to narrow down the pool of subjects in the research of missing persons and, eventually, to assist in identifying unknown individuals. In forensic anthropology, the four major pillars of the biological profile are skeletal sex, ancestry, age at death and stature. Skeletal sex estimation relies on determining morphological and metric dimorphic features within the pelvis [1,2], the cranium [3,4] and long bones [5]. Ancestry estimation is mainly based on the evaluation of morphoscopic [6,7,8] and metric cranial traits [9]. These two parameters can be inferred both by osteological and genetic evidence. Currently, forensic DNA typing uses STR markers to establish the identity of unidentified remains, link a person to a crime scene [10] and confirm familial relations [11]. In some cases, Y-chromosome and mitochondrial-DNA (mtDNA) analysis can be used to integrate autosomal STR analysis. However, neither of these two markers alone can identify a person because multiple individuals in any given population can have the same Y-chromosome/mtDNA profile. Still, both can either rule out matches or increase the significance of a match [12].

Obtaining a DNA profile to identify human remains is of limited use if antemortem data such as a direct reference sample (biopsy and other archived medical samples belonging to the victim) or a biological sample of the victim’s relatives (parents, children, and full siblings) are not available for comparison. In such circumstances, as occurs, for example, in armed conflicts, other situations of violence or unidentified persons, obtaining additional information—such as, for instance, biogeographical ancestry (BGA)—from DNA analysis could be beneficial in providing support in the creation of the biological profile, directing investigative activity in search of family members of the victims, and assisting with the identification of unknown remains. Therefore, in this paper, a multidisciplinary approach (anthropological and molecular) to infer the sex and biogeographical ancestry of different skeletons using routine forensic STR markers, such as autosomal and Y STR and mitochondrial profile, was explored. Recently, Yang et al. [13] also tested the possibility of inferring facial characteristics through the analysis of 15 STRs and highlighted the difficulty of retrieving phenotypic information by analyzing STRs loci. In addition, Alladio et al. [14] defined an approach to estimate the likelihood ratio for BGA purposes involving multivariate-data-analysis strategies for assessing biogeographical ancestry. Powerful multivariate techniques such as principal component analysis (PCA), partial least squares-discriminant analysis (PLS-DA), and support vector machines (SVM) have been used and compared. In particular, PLS-DA proved to be a robust classifier and yielded models with high sensitivity and specificity, capable of discriminating populations on a BGA basis.

Here, the comparison of physical, genetic, and then antemortem data (once the individual had been identified) was particularly beneficial to evaluate the accuracy rates of the biological profiles produced by anthropologists and classification rates obtained by molecular experts using autosomal genetic profiles and multivariate statistical approaches. Thomas et al. carried out two retrospective studies on accuracy rates for skeletal sex [15] and ancestry [16], collecting casework from the FBI Laboratory. Over 90% of the correct classification rates were recorded both for sex and ancestry. To date, no retrospective study has been carried out by molecular experts to predict the accuracy of classification methods such as PLS-DA when used in association with autosomal genetic profiles. Therefore, in this paper, we decided to present a retrospective study on a pool of 24 forensic cases of skeletal remains that underwent autopsies at the Institute of Legal Medicine of Milan. The aim of the paper is threefold: (i) to investigate the consistency of the observations between physical and molecular analyses for sex and ancestry estimation; (ii) to compare anthropological/molecular data to antemortem data (where possible) and evaluate both the anthropological and genetic correct classification rates; (iii) to discuss if the genetic markers that are routinely used in forensic cases for identification purposes can also be considered valuable allies in inferring sex and biogeographical ancestry, as well.

## 2. Materials and Methods

### 2.1. Sample

The specimens selected were from 24 forensic cases of human skeletal remains investigated by the Laboratorio di Antropologia e Odontologia Forense (LABANOF) of the University of Milano between 1997 and 2018 (Table 1). The remains arrived as unidentified individuals and were stored at the laboratory facilities while waiting for anthropological analyses [17]. Among the bone samples available, femora, tibiae and petrous bones were selected from each case for genetic analyses, so that 24 skeletal samples were collected. Following investigations, nine out of twenty-four cases were positively identified. Antemortem data concerning sex and biogeographical ancestry were recorded and used here as positive controls, although this applies only to this subsample of identified individuals. The skeletons or skeletal remains presented differently, with various decomposition stages and preservation states (from partial to extensive or complete skeletonization, adipocere, burnt). Postmortem interval (PMI) at recovery ranged from a few weeks to 20 years. Most frequently, the skeletons were remarkably well preserved in terms of quantity and quality of the bone. Depending on the conditions, each skeleton was treated differently to preserve the remains best. However, the general preparation consisted of brushing the remains in lukewarm water to remove dirt and soft tissue remnants. Fragile specimens, such as burnt remains, were delicately groomed with a soft, moistened brush.

### 2.2. Physical Analyses

For each case, a biological profile was created. Since this is a retrospective study, different methods were used according to the time of the analysis. In general, sex estimation was based on morphological traits of the cranium [18] and of the pubic bones [2]. Ancestry was estimated by assessing morphoscopic traits of the cranium, such as palatal morphology, maxillary projection (prognathism), nasal opening, nasal sill/guttering, nasal spine, nasal bridge form, and incisor form [6,8]. In some cases, osteometric and craniometric analyses were carried out using the discriminant function program Fordisc 3.0 to estimate ancestry and sex [9]. Age at death was estimated according to the degenerative changes in skeletal districts, such as the pubic symphysis [19], the sternal end of the fourth rib [20,21], the auricular surface [22], and in dentition [23,24].

### 2.3. Molecular Analyses

Molecular analyses were performed in two different laboratories: Parma Forensic Biological unit of Carabinieri and Forensic molecular anthropology unit of University of Florence.

#### 2.3.1. Parma Forensic Biological Unit of Carabinieri

Sample Inspection and Processing

At first, each piece of evidence was roughly shattered, then externally cleaned to allow the selection and collection of a small bone fragment, and subjected to further decalcification before turning it to dust and proceeding with the forensic genetic inspections.

Fragments of about 400 mg to 500 mg from each bone sample were treated with 1.5 mL of 0.5 M EDTA and incubated at 37 °C for at least 3 to 4 days until the bone was completely decalcified [25,26].

A second overnight incubation step at 56 °C was then applied to the fully decalcified bone fragments after adding 100 μL of G2 lysis Buffer (Qiagen, Hilden, The Netherlands), 60 μL of Proteinase K (Qiagen) and 20 μL of dithiothreitol (Qiagen). The samples’ nucleic acids were then extracted with the EZ1^®^ DNA Investigator kit (Qiagen, 2017), according to the manufacturer’s instructions for the “trace protocol”, with a final elution in TE buffer set at 50 µL volume.

The extracted samples were quantified using the Quantifiler™ Trio DNA Quantification kit (Thermo Fisher Scientific, Waltham, MA, USA, 2017) run on the Applied Biosystem^®^ 7500 Real-Time PCR System and analyzed with the HID Real-Time PCR Analysis Software v 1.3.

STR Loci Amplification: Autosomal DNA and Y Chromosome Markers

The extracted and quantified samples were then subjected to STR marker amplification, based on multiplex PCR systems available on the market: PowerPlex^®^ Fusion 6C System (Promega, Madison, WI, USA) was used for the autosomal STR loci analysis, and the PowerPlex^®^ Y23 System (Promega) for the Y Chromosome STRs.

Typing of Both Autosomal and Y-Chromosome Markers

For typing the amplified genetic material, the capillary electrophoresis (CE) technique [27] was used, run on the Applied Biosystem^®^ Genetic Analyzer 3500 XL (Thermo Fisher Scientific, Waltham, MA, USA, 2009), with the proper reagents and the suggested parameters as indicated by the manufacturer Promega for both the Powerplex^®^ Fusion 6C System and the PowerPlex^®^ Y23 System; data interpretation was supported by the analysis conducted with the GeneMapper^®^ ID-X software v. 1.4 (Thermo Fisher Scientific, Waltham, MA, USA, 2009). The Y-chromosome haplogroup inference was determined with calculations performed on the free online tool NEVGEN Y-DNA Haplogroup Predictor (http://www.nevgen.org, accessed on 4 May 2023), using general level for prediction.

#### 2.3.2. Forensic Molecular Anthropology Laboratory (Florence)

Sample Preparation and DNA Extraction

Bone samples were processed in the molecular anthropology unit of the University of Florence, a state-of-the-art facility dedicated to the analysis of degraded DNA samples.

The outer layer of the bones was mechanically removed to remove potential contamination using a rotary sanding tool (Dremel^®^ 300 series). After brushing, each sample was irradiated with ultraviolet light for 45 min in a Biolink DNA Crosslinker (Biometra, Göttingen, Germany). A minimally invasive approach was followed to recover approximately 50 mg of bone powder from petrous bones, as described by Sirak et al. [28]. DNA was extracted using silica-based protocol [29] and DNA was eluted in TET buffer (10 nM Tris, 1 mM EDTA, 0.05% Tween-20) twice for a final volume of 100 μL. DNA was quantified using Qubit^TM^ 4 Fluorometer (dsDNA High Sensitivity Kit).

Library Preparation, Enrichment and Data Processing

A double-stranded and dual-indexed library for each sample was constructed starting from 20 µL of extracts [30]. MtDNA molecules were captured following the protocol described in Maricic et al. [31] and sequenced on an Illumina MiSeq run for 2 × 76 + 8 + 8 cycles. After demultiplexing, the sequences were sorted according to the individual sample barcodes and processed using the pipeline described by Peltzer et al. [32]. Clip & Merge v1.7.4 was used to trim adapters and merge the reads with a minimum overlap of 10 bp in a single sequence. Merged reads were mapped on the revised Cambridge Reference Sequence (rCRS, NC_012920) by CircularMapper and BWA v.0.6.2 [33], setting parameters proposed by Zimek et al. [34]. Reads with mapping quality below 30 were discarded. Then, DeDup v0.12.1 was used to remove PCR duplicates and degradation patterns were estimated using MapDamage 2.0 [35]. Schmutzi software [36] was used to evaluate contamination levels and obtain the final endogenous consensus profile. Bases with individual likelihood <30 were considered as missing positions (Ns) in the reconstructed mitogenome sequences. Haplogrep3 [37] was selected to determine the mitochondrial haplogroup for each sample. Blank controls were included during all experimental steps.

Statistical Analysis

The PLS-DA method, as described in the study by Ruiz-Perez et al. [38], aims to identify the variables that explain the majority of the variation in predictor variables while simultaneously modeling the predictors that are most strongly correlated with response variables (biogeographical ancestry, in this case). The PLS algorithm identifies the first component, or latent variable (LV), that explains most of the variation in the response variable, and subsequent components explain the remaining variation [39]. The regression line slopes, or PLS weights, indicate the direction of the first LV. Unlike other regression methods, PLS can tolerate noisy or redundant data, and the data that do not fit the model are considered residuals [40]. In contrast, LDA is a supervised classification method that distinguishes different class objects by evaluating their optimal boundaries. It allows the distinguishing of samples of different categories by examining the probability distributions of the categories to which the samples might belong. Accordingly, each sample is placed in the category that has the highest value in terms of probability. Graphically, the probability distributions are represented as ellipses at different probability levels for each class examined. These ellipses are each tangent to a point halfway between the class centers. A straight-line delimiter serves as a boundary to separate the ellipses and, thus, the different categories. LDA provides a linear function of the variables and maximizes the ratio between the variances of each class. The weights are chosen to provide the best classification of the objects, allowing LDA to select the direction that achieves the maximum separation between the given categories [41]. For this study, PLS-DA models were constructed on STR, Y haplotype, and mitochondrial markers using repeated double cross-validation procedures according to Varmuza et al. [42], which involved a venetian-blind sampling design and k-fold equal to 5.

The autosomal STRs dataset for PLS-DA analysis was composed of 799 individuals (361 = Caucasian—the term has been replaced throughout the text with European American—, 97 = Asian and 341 = African-American) from the NIST 1036 Revised U. S. Population Dataset [43]. The Y-STRs dataset [44] was composed of 19,494 individuals (3651 = Asian, 1327 = African, 13227 = European, 733 = Mixed American and 556 = Native American) and mitochondrial dataset from HmtDB [45] was composed of 25218 individuals (6021 = Asia, 3762 = Africa, 11421 = Europa, 2984 = America and 1030 = Oceania).

Accuracy, sensitivity, and specificity metrics were estimated for all the developed PLS-DA models. Once developed, the models were tested to predict 24 samples for each marker under exam, i.e., STRs, Y STRs and mitochondrial markers.

## 3. Result and Discussion

### 3.1. Physical Results

Skeletal sex and ancestry estimations classified the samples as follows: 13 European male individuals and 7 European and 3 African females. Evidence 10 was a human cranium, whose dimorphic traits were too ambiguous to express a conclusive sex estimation (hence, indeterminate), whereas the estimated ancestry was European (Table 2). Age at death is shown to complete the biological profile, but it does not fall within the scope of the study as it cannot be determined with genetic analyses.

### 3.2. Molecular Results

#### 3.2.1. Autosomal and Y STRs

Autosomal STR typing was performed on all extracted samples, and complete profiles were obtained from 15 out of 24 samples (samples 1, 2, 3, 4, 9, 10, 12, 16, 18, 21, 22, 23, 24, 25 and 26). While partial profiles with a number of loci typed ≥13 were obtained from six samples (samples 6, 7, 8, 14, 15 and 20). There were no results from 3 out of 24 samples (sample 5, for which only the sex was identified, 17 and 19). STR typing highlighted that 12 out of 24 skeletal remains belonged to individuals of the male sex and 10 were females. PLS-DA was performed on all STR profiles except for the three samples from which no results were obtained (Appendix A). In addition, predictive analysis was performed for each of the samples with partial profiles by fitting the database to the individual sample to obtain a more accurate result.

Y STR typing was performed on 12 male individuals and complete profiles were obtained from 7 out of 12 samples (samples 1, 2 (only one locus was missing), 3, 4, 9, 21, and 24). Partial profiles with the number of loci typed being 17, 21 and 16 out of 23 were obtained from samples 12, 16 and 26, respectively, and 9 loci were typed from sample 5. No profile was achieved from sample 8. The Y haplogroup of each sample was assigned using the Nevgen Y-DNA Haplogroup Predictor tool and the geographical distribution is reported in Table 2. As can be observed in the table (Table 2), no Y haplogroup was assigned to sample 5 and 8, and, consequently, it was impossible to make any biogeographical prediction. PLS-DA was also performed on all Y STR profiles except for sample 8, from which no results were obtained (Appendix A). Also in this analysis, the database was adapted for both samples with partial profiles and Evidence 2, as the DYS389II *locus* was missing.

#### 3.2.2. Mitochondrial DNA

MtDNA analysis was performed on all extracted samples, and complete profiles were obtained from 24 out of 24 samples. The 24 mitogenomes were sequenced to an average coverage depth of 507.40X (from 103.40X to 979.72X) (Appendix A).

In addition, Appendix A shows the results obtained for estimating current human contamination. As proposed by Renaud et al. [36], the endogenous consensus was accurately reconstructed for all samples with the exception of Evidence 9 and 26, which showed high levels of contamination. Sample cleanup using the PDMtools software (https://github.com/pontussk/PMDtools, accessed on 16 March 2022) was subsequently performed and the new consensus was used to estimate the haplogroup. Sequences were assigned to eight distinct haplogroups, as seen in table (Table 2).

### 3.3. PLS-DA Results

#### 3.3.1. Autosomal STRs

The results obtained from the PLS-DA analysis for autosomal STRs are shown in Appendix A. A prediction value was associated with each sample for each of the three continents. Observing the highest prediction values, it was possible to conclude that evidence 22 fell among Asian individuals, evidence 2, 7 and 23 among African-American individuals, while the remaining evidence fell among European-American individuals. The PLS-DA analysis plot (Figure 1) shows how complete profiles of unknown samples are distributed among the 799 individuals in the dataset. Since there were different numbers of markers for evidence 6, 7, 8, 14 and 20, the PLS-DA model was adapted to every sample by forming five plots. No prediction was obtained for evidence 15, probably due to a low number of loci.

#### 3.3.2. Y STRs

The result obtained from the PLS-DA method for Y-STRs can be seen in Appendix A. A prediction value was associated with each sample for the five continents. Observing the highest prediction values, it is possible to conclude that all evidence falls among European individuals. No prediction was obtained for evidence 5, probably due to the presence of a low number of loci. The PLS-DA analysis plot (Figure 2) shows how our unknown samples are distributed among the 19494 individuals in the dataset. For evidence 2, 12, 16 and 26, which have different numbers of markers, the PLS-DA model was adapted to every sample by forming four plots.

#### 3.3.3. mtDNA

The result obtained from the PLS-DA method for mitochondrial DNA can be observed in Appendix A. A prediction value was associated with each sample for each of the five continents. Observing the highest prediction values, it was possible to conclude that evidence 1, 2, 3, 4, 5, 8, 9, 16, 17, 18, 19, 20, 21, 22, 24, 25 and 26 fall among European individuals, evidence 6 and 14 among Asian individuals, evidence 7, 12 and 23 among African individuals. For evidence 10 and 15, we obtained no predictions. The PLS-DA analysis plot (Figure 3) shows us how our unknown samples are distributed among the 25218 individuals in the dataset.

### 3.4. Mirroring Physical and Genetic Results

This paper presented a collation of 24 cases of unidentified bodies in Milan where physical and molecular anthropologists worked together to estimate two pillars of the biological profile, namely, skeletal sex and ancestry. The results of both analyses are summarized in Table 3. As for sex estimation, the disciplines are in perfect agreement, even though physical or molecular analyses were inconclusive in some instances (i.e., evidence 10, 17 and 19). Evidence 10 was an isolated human cranium presenting ambiguous traits for which physical methods could not produce a conclusive sex estimation (hence, it was considered indeterminate), whereas DNA analysis confirmed that it was a female individual. On the contrary, DNA typing did not produce exhaustive profiles for evidence 17 and 19, for which only skeletal-sex estimation is available. Such an agreement was not observed in ancestry estimation, as shown by evidence 2, 6, 12, 14 and 22, where physical and molecular estimations results were partially inconsistent both between the two methods used and the different genetic markers investigated. As for molecular analysis, the observed inconsistencies among genetic markers could be explained with the nature of the markers used in the study—STRs are not the best choice to infer biogeographical ancestry— the database and/or the presence of admixed ancestry. As proposed in Alladio et al. [46] and Pilli et al. [47], particular attention should be paid to the database. Since the analysis of ancestry inference is performed by comparing the sample genotype with one or more known reference population groups, well-characterized databases with high-quality genotyping results of well-defined reference populations are critical. This discrepancy raises two issues: (i) can wide physical and molecular classifications be considered an entirely robust and conclusive proxy to infer the actual provenance of an individual? (ii) What discipline is more informative for an individual’s ancestry, when they provide partial, inconsistent results and routine forensic STR markers were used? (iii) How can the discrepancy observed in the genetic markers be interpreted? Which marker could be more reliable than others?

To attempt to answer these questions, a subset of subjects that were identified during our research activity was evaluated. The subset of identified subjects was used to compare the results of the physical and molecular pipelines to known sex and provenance following positive identification (Table 4). Both physical and molecular sex estimations are perfectly in line with the antemortem data, so that the accuracy rate of sex estimation against known sex is at 100% in this subset. As for ancestry, physical methods classified the remains into macro groups that can be considered to some extent consistent with the known provenance. Molecular analysis also classified the remains into macro groups consistent with the physical findings and known ancestry for evidence 1, 7, 8, 9, 25 and 26 (66.6% was the percentage of success in ancestry estimation). However, molecular analysis does not return concordant results in the markers used in 33.3% of the cases (evidence 2, 6 and 12 Table 4) unless the presence of admixed ancestry is considered. For example, evidence 12 was estimated as European by physical methods and STRs analysis of both autosomal and Y loci, although the actual provenance was Morocco (marked in orange in Table 4). The unique marker that correctly classified the individual as African was the mtDNA. The partial discordance in assigning biogeographical ancestry in some samples could be explained by the presence of admixed ancestry or an unrepresentative reference population, which entails that physical anthropological classification into binding groups needs further interpretation [48]. However, this aspect was not investigated since the known provenance was based on antemortem data of the identified individuals, including place of birth, but no information on admixed ancestry was available.

Particular attention should be paid to the PLS-DA prediction value of autosomal and Y markers (Appendix A). In fact, in all cases in which the prediction value was higher than 0.8, the inference to a specific biogeographical area was performed correctly except for evidence 26, in which the autosomal STR prediction value of 0.64 allowed us to correctly assign the individual to the European-American population, and evidence 9, which was properly assigned to the European population with prediction value of 0.49.

STR markers are currently used for human identification in most forensic laboratories because they are highly polymorphic and unique to each individual, except for identical twins. Due to their high variability, STR markers can match DNA samples from different sources and identify individuals with a high degree of certainty, even from very small or degraded DNA samples. Even if, sometimes, STRs can be used for studying genetic diversity and relationships among populations using supervised machine-learning approaches (as reported, for instance, in Forensic Sci Int Genet. [14]), they have not been shown to be as accurate for inferring ancestry as other genetic markers such as single-nucleotide polymorphisms (SNPs) [49]. This concept is closely connected with the fact that the information content of STRs is relatively low, meaning that the number of alleles (versions) at each STR locus is limited, and the pattern of variation may not reflect population history as accurately as other genetic markers. In addition, the inheritance patterns of STRs can be complex. Furthermore, STRs recovered on forensic evidence may be influenced by genetic mutations and degradation, making it challenging to infer ancestry based on a small set of STR markers alone. Therefore, while STR markers can be helpful for individual identification, they are not as well suited for inferring biogeographical ancestry as other genetic markers, such as SNPs or mitochondrial DNA. In addition, the choice of the database is also crucial to properly infer the biogeographical ancestry of unknown individuals via supervised/classification modeling techniques.

In conclusion, this study extended the research line of comparative studies on physical and DNA evidence for the sex and ancestry estimation of skeletal remains from casework. As seen here, physical and molecular sex estimations are confirmed to be highly reliable, with higher accuracy rates (100%) than those previously reported (94.7%) [15]. It is to be noted that the sample is here limited to the cases held by the LABANOF facilities, whereas previous research [15,16] considered broader settings (e.g., the FBI database). Such an agreement was not observed when comparing physical and molecular data of the ancestry analyses. A previous paper assessed an overall 90.9% accuracy rate of estimated versus identified ancestry [16]. However, our study expanded this research line by adding genetic ancestry data and it started investigating the potential drawbacks that still hinder the elaboration of conclusive results for ancestry estimation.

## 4. Conclusions

Physical and molecular anthropology have been making advances in the field of forensic science and can now contribute to identifying missing persons by providing information about skeletal sex and ancestry through physical and molecular methods. Therefore, the present study evaluated the sex and biogeographical ancestry of different skeletons using physical methods and routine forensic STR markers associated with multivariate statistical approaches. A recent paper reported the challenges faced by anthropologists when studying a migrant population via SNP markers and the NGS (next-generation sequencing) approach, where such an agreement between disciplines was not observed [50]. Here, the possibility to compare physical/molecular with antemortem data (some individuals were identified during our research) allowed us to evaluate both the anthropological and genetic correct classification rates, even if only on limited number of samples (nine samples). The results suggested a perfect match between physical and molecular data and an outstanding classification rate in sex estimation for both disciplines when the results were compared with antemortem data. A different picture was observed when the ancestry estimation was evaluated. The comparison between physical and molecular data showed a discrepancy in 5 out of 24 samples but the subsequent comparison between physical/molecular and antemortem data on nine samples showed that physical methods failed only once in the ancestry estimation (evidence 12 was classified as a European whereas he was from Morocco) while molecular ones failed three times, with discrepancies also among the different markers used. Particular attention should be paid when using STR markers to infer biogeographical ancestry. In fact, if, on one hand, the use of the STR profile to also predict the biogeographical ancestry could save money, time, and sometimes precious biological materials—such as, for example, the DNA extract from micro traces collected at a crime scene—, on the other hand, a loss in inference accuracy could be observed since STR markers did not appear to be reliable when inferring biogeographical ancestry, despite the use of supervised machine-learning approaches. Further studies with more antemortem data will be needed to support the results obtained. To conclude, it is important to keep in mind that a well-characterized database with high-quality genotypes of well-defined reference populations is crucial to accurately infer biogeographical ancestry for both physical and molecular methods.

## Figures and Tables

**Figure 1 genes-14-01064-f001:**
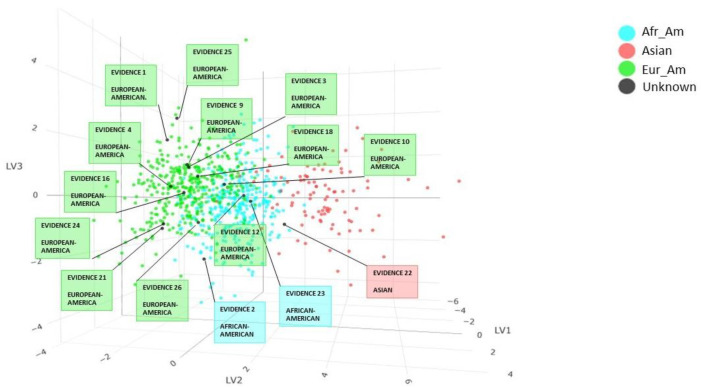
Individuals from the database belonging to the different continents are represented with different colors. Unknown individuals are in black, and each black dot has been labeled with the specimen name and the prediction obtained from the autosomal STRs PLS-DA model. Score plot LV1, LV2 and LV3.

**Figure 2 genes-14-01064-f002:**
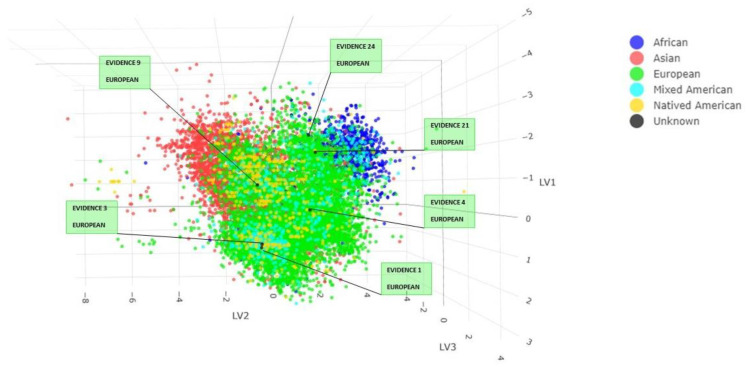
Individuals from the database belonging to the different continents were represented with different colors. Unknown individuals are in black, and each black dot has been labeled with the specimen name and the prediction obtained from the Y-STRs PLS-DA model. Score plot LV1, LV2 and LV3.

**Figure 3 genes-14-01064-f003:**
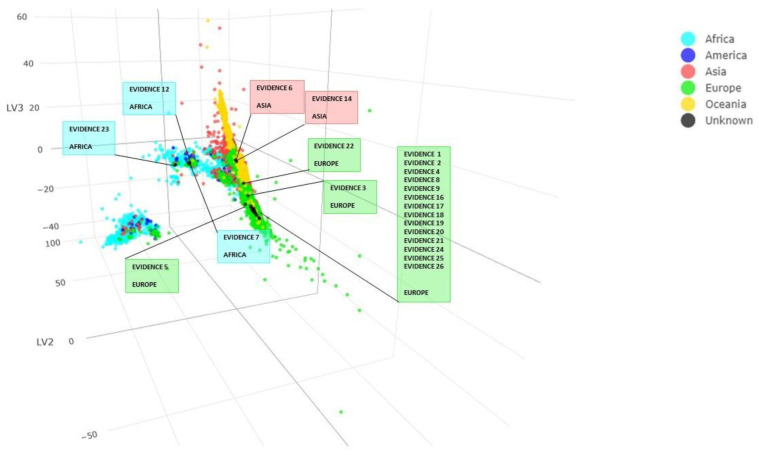
Individuals from the database belonging to the different continents are represented with different colors. Unknown individuals are in black, and each black dot has been labeled with the specimen name and the prediction obtained from the mitochondrial PLS-DA model. Score plot LV1, LV2 and LV3.

**Table 1 genes-14-01064-t001:** List of cases used for the study. For each case, the taphonomic condition at the time of recovery and the PMI (postmortem interval) are reported.

Case ID	Taphonomic Condition	PMI at Recovery	Bone Sampled for Genetics
Evidence 1	Skeletonized	Few months	Tibia
Evidence 2	Skeletonized	10 years	Femur
Evidence 3	Skeletonized	17 years	Femur
Evidence 4	Skeletonized	17 years	Femur
Evidence 5	Skeletonized	1–2 years	Femur
Evidence 6	Partially skeletonized, burnt	Few weeks	Tibia
Evidence 7	Skeletonized	Few months	Femur
Evidence 8	Skeletonized	7 years	Petrous bone
Evidence 9	Partially skeletonized, burnt	1 week	Femur
Evidence 10	Skeletonized	20 years	Petrous bone
Evidence 12	Skeletonized	8 years	Femur
Evidence 14	Skeletonized	6–12 months	Femur
Evidence 15	Skeletonized	Few months	Femur
Evidence 16	Skeletonized	1–2 years	Femur
Evidence 17	Adipocere, skeletonized	6–12 months	Femur
Evidence 18	Skeletonized	8 months	Petrous bone
Evidence 19	Skeletonized	3–6 months	Femur
Evidence 20	Skeletonized	3–6 months	Femur
Evidence 21	Skeletonized	5–6 months	Femur
Evidence 22	Skeletonized	3 months–1 year	Petrous bone
Evidence 23	Skeletonized	1–2 years	Tibia
Evidence 24	Skeletonized, partially burnt	3–7 months	Femur
Evidence 25	Partially skeletonized	1–2 weeks	Femur
Evidence 26	Skeletonized	1–3 years	Femur

**Table 2 genes-14-01064-t002:** In the first four columns are case ID and results of the sex and ancestry estimation based on physical evidence. IND: indeterminate. Then, biogeographical prediction for the Y chromosome haplogroups and probability assigned by Nevgen tool. NA: not available; ND: not detectable. In the latter two columns, haplogroup assigned by Haplogrep3 software and worldwide distribution of haplogroups based on the continent with the highest frequency to which that specific haplogroup was assigned.

Case ID	Skeletal Sex	Skeletal Ancestry	Age-at-Death Estimation (Years)	Y-DNA Haplogroup Probability Assigned by Nevgen Predictor	mtDNA Haplogroup	Continent
Evidence 1	M	European	43–57	R1b M269—western Europe 100%	H1(H1)	Europe
Evidence 2	M	European	40–54	R1b M269—western Europe 100%	U5a(U5a1b1g)	Europe
Evidence 3	M	European	21–56	R1b M269—western Europe 99.92%	R0(R0)	South Asian
Evidence 4	M	European	26–45	I2a1b3—south-eastern, south-western, north-western Europe 100%	H(H)	Europe
Evidence 5	M	European	26–45	ND	U6a(U6a3b)	Europe
Evidence 6	F	European	30–50	Female	M5a(M5a1b)	South Asia
Evidence 7	F	African	18–22	Female	L2a(L2a1a1)	Africa/Africa America
Evidence 8	M	European	>60	NA	U4b(U4b1a3a)	Europe
Evidence 9	M	European	36–50	J2a1 M67—Europe, the Middle East and northern Africa 96.4%	T2e(T2e2a)	Europe
Evidence 10	IND	European	18–39	Female	L3e(L3e1a3a)	Africa/Africa America
Evidence 12	M	European	35–44	E1b1b L67–Europe, the Near East, and northern Africa 100%	L2a(L2a1c1)	Africa/Africa America
Evidence 14	F	European	38–50	Female	M5a(M5a1b)	South Asia
Evidence 15	F	European	34–63	Female	M1a(M1a1)	South Asia
Evidence 16	M	European	30–44	I2a1b3—south-eastern, south-western, north-western Europe 100%	T1a(T1a10)	Europe
Evidence 17	M	European	57–71	NA	H1b(H1bp)	Europe
Evidence 18	F	European	17–22	Female	K1a(K1a)	Europe
Evidence 19	F	European	30–50	NA	T2(T2+16189)	Europe
Evidence 20	F	European	17–22	Female	H41a(H41a)	Europe
Evidence 21	M	European	32–52	E1b1b >V13—Europe, the Near East, and northern Africa 99.92%	H1b(H1b1+16362)	Europe
Evidence 22	F	African	28–52	Female	R9c(R9c1b1)	South Asia
Evidence 23	F	African	37–52	Female	L2b(L2b1b)	Africa/ Africa America
Evidence 24	M	European	60–80	E1b1b >V13—Europe, the Near East, and northern Africa 74.5%	U5a(U5a1c)	Europe
Evidence 25	F	European	39–53	Female	J2b(J2b1c)	Europe
Evidence 26	M	European	>60	R1b—western Europe	K1a(K1a4a1h)	Europe

**Table 3 genes-14-01064-t003:** Results of the physical and genetic analyses for the creation of the biological profile of the 24 cases. IND: indeterminate; NA: not available; ND: not detectable. The discrepancies observed between physical and molecular analysis and among genetic markers are marked in red. Eur_Am: European American; EU: European; Afr_Am: African American; AS: Asian; AF: African.

Case ID	Physical Sex	Molecular Sex	Physical Ancestry	Genetic Ancestry		
				STR	Y	MtDNA
Evidence 1	M	M	European	Eur_Am	EU	EU
Evidence 2	M	M	European	Afr_Am	EU	EU
Evidence 3	M	M	European	Eur_Am	EU	EU
Evidence 4	M	M	European	Eur_Am	EU	EU
Evidence 5	M	M	European	NA	ND	EU
Evidence 6	F	F	European	Eur_Am	NA	AS
Evidence 7	F	F	African	Afr_Am	NA	AF
Evidence 8	M	M	European	Eur_Am	NA	EU
Evidence 9	M	M	European	Eur_Am	EU	EU
Evidence 10	IND	F	European	Eur_Am	NA	ND
Evidence 12	M	M	European	Eur_Am	EU	AF
Evidence 14	F	F	European	Eur_Am	NA	AS
Evidence 15	F	F	European	ND	NA	ND
Evidence 16	M	M	European	Eur_Am	EU	EU
Evidence 17	M	ND	European	NA	NA	EU
Evidence 18	F	F	European	Eur_Am	NA	EU
Evidence 19	F	ND	European	NA	NA	EU
Evidence 20	F	F	European	Eur_Am	NA	EU
Evidence 21	M	M	European	Eur_Am	EU	EU
Evidence 22	F	F	African	Asian	NA	EU
Evidence 23	F	F	African	Afr_Am	NA	AF
Evidence 24	M	M	European	Eur_Am	EU	EU
Evidence 25	F	F	European	Eur_Am	NA	EU
Evidence 26	M	M	European	Eur_Am	EU	EU

**Table 4 genes-14-01064-t004:** Comparison of results of the physical and genetic analyses for the creation of the biological profile of the nine identified cases. Known sex and provenance indicate the antemortem data upon completion of positive identification. The discrepancies observed between physical and molecular analysis and among genetic markers are marked in red and orange. NA: not available. Eur_Am: European American; EU: European; Afr_Am: African American; AS: Asian; AF: African.

Case ID	Physical Sex	Molecular Sex	Known Sex	Known Provenance	Physical Ancestry	Genetic Ancestry		
						STR	Y	mtDNA
Evidence 1	M	M	M	Italy	European	Eur_Am	EU	EU
Evidence 2	M	M	M	Italy	European	Afr_Am	EU	EU
Evidence 6	F	F	F	Ukraine	European	Eur_Am	NA	AS
Evidence 7	F	F	F	Nigeria	African	Afr_Am	NA	AF
Evidence 8	M	M	M	Germany	European	Eur_Am	NA	EU
Evidence 9	M	M	M	Italy	European	Eur_Am	EU	EU
Evidence 12	M	M	M	Morocco	European	Eur_Am	EU	AF
Evidence 25	F	F	F	Italy	European	Eur_Am	NA	EU
Evidence 26	M	M	M	Italy	European	Eur_Am	EU	EU

## Data Availability

The data presented in this study are available upon request.

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
