# Peer review of "Comparing Genetic and Physical Anthropological Analyses for the Biological Profile of Unidentified and Identified Bodies in Milan"

_genes, 2023, doi:10.3390/genes14051064_

Round 1

Reviewer 1 Report

Dear Authors,

this paper is very interesting and I believe that it gives a new insight into two methodologies in real-life cases. 

Taking that into consideration I think that this is more a case study; the sample size is to small to make generalized conclusions. This should be changed in the conclusions part. 

Additional issues that you should correct are: 

Line 43. You are missing a full stop after reference 5.

Line 209. You are missing a space between the word and reference 42. 

You have to unify writing of PLS - DA in text (you have different spacing). 

Table 3. Line 256; you have two full stops.

Tables 2, 3, 4 would be better combined; thus all of the data could be more transparent and visible. 

Author Response

Thank you

Stefania Morelli 

Reviewer 2 Report

The authors present a study in which routinely tested forensic markers, namely aSTRs, YSTRs and mtDNA are used to interfere biogeographic ancestry of unknown human remains collected during missing person investigations. Even though the idea is not novel, the forensic community might find it interesting to learn how informative the golden forensic standards are when combined and analyzed with specific statistical methods.

The submitted manuscript requires some revisions before accepted:

- the Y-lineage prediction from Nevgen - please reference at least the website, not all the readers might be familiar with the tool

- please report the predicted Y-lineage with the probability given by Nevgen. also, I suggest reporting not only the major haplogroup but also the most derived predicted by Nevgen (with % too)

- please include in the table how many Y-STRs were used for the prediction - the probability of incorrect prediction for incomplete profiles is higher

- please review the Table 4 as it seems to contain mistakes - e.g. Evidence 10 is assigned to mtHG L3e(L3e1a3a) which is described as south European

- the authors discuss PLS-DA results and among them, Evidence 7, which is not presented on the Figure 1 to which they refer

- I would argue with stating that Evidence 12 is wrongly assigned by anthropology - North African population cannot be compared to Sub-Saharan Africa. Generally, I don't think that referring to the 'race' as seen by anthropologists should be part of the manuscript

- Table S2 - status NA was given to Evidence 8 (failed) and to female samples - please change as this is confusing. when reviewing the data if makes a difference if the sample was male and the analysis failed or if the sample just had no Y chromosome

- how would authors explain no predictions for Evidence 10 and 15 for the mitochondrial DNA if the whole mitogenomes were used?

What was the known biogeographic ancestry based on? Place of birth only? This has to be clarified as this level of information does not exclude possible co-ancestry reflected by genetic results. The authors do not address through the manuscript a possibility that the studied individuals were of admixed ancestry, which would of course result in not consistent classification of haploid markers and also affect probabilities obtained for the diploid markers. I would suggest not to refer to the anthropological results, rather expand the genetic data interpretation.

Author Response

Thank you

Stefania Morelli

Reviewer 3 Report

This article is interesting because it aims to compare anthropological and genetic methods used in current forensic practice for individual identification, regarding the determination of sex and biogeographical ancestry.

However, anthropological methods are insufficiently described and for the BGA question little or poorly referenced. The reference database for autosomal genetic polymorphism STRs is based on obsolete classification terminology, which needs to be rectified.

These points can easily be improved.

line 65: acronym BGA : define before using it, for instance on line 56 : biogeographical ancestry (BGA)

line 71: “capable of discriminating population on an ethnic basis”. What does it mean? Ethnicity is defined on a cultural basis, not on a biological one (see Max Weber’s work). As this article is about phenotypic and genotypic variability, and not cultural diversity, it is more appropriate to use the expression “discriminating population on an BGA basis” as it is done in the rest of the article.

line 102: PMI ranged from a few months to 20 years, but in table 1 some evidence (9, 24, 25) are ‘younger’, the youngest is evidence 24: 3-7 days.

line 114-115: authors wrote: “Ancestry was estimated using morphoscopic traits of the cranium [6, 18]”.  Please list these cranial morphoscopic traits and the methods used for assessing BGA from the skull morphology. The issue is that in the reference 6 quoted (Hefner’s paper), one can read in its conclusions: “On the contrary, a combination of supposedly ancestrally diagnostic traits in many individuals shows the fallacy of the typological approach to ancestry prediction and reveals variation in morphoscopic traits within ancestral groups.” Consequently Hefner recommended the use of statistical methods for an optimal  weighing of the traits used to predict BGA on a given skull. Moreover the ref 18 that is quoted : Buikstra & Ubelaker, 1994 (by the way, please correct and add the full name of the latter author in the reference list) do no mention any cranial traits for determining BGA. Please pay attention to these two issues, which are crucial for the purpose of this paper.

line 141: edit “amplificantion”

line 214-218 : for autosomal STRs dataset, the reference sample contains the group “Caucasian”, whereas for Y-STRs and mt dataset, “European” and “Europa” are respectively mentioned as population subgroups of the general reference sample.

It's quite strange... In the United States the population of European origin would come only from the Caucasus region? It seemed to me that Great Britain, Ireland and other Western European countries had contributed more to the settlement of the United States of America than the countries of the Caucasus... More seriously, the term "Caucasian" is a totally obsolete term in scientific language in general and anthropological vocabulary in particular to designate European populations or of European ancestry. This term was introduced in the 18th century by Blumenbach and it served later a till the end of the WW2 a typological and raciological discourse which has no scientific validity. It should be banned from the scientific literature of the 21st century. If it is still used in the USA, it is essentially for sociological and political reasons and many American scholars ask for its suppression from scientific studies. It is quite revealing to note here that it is the reference concerning the USA, produced by American researchers, which uses this term for the analysis of the autosomal STRs polymorphism in the North American population, whereas the studies of the Y-STRs and mt-STRs polymorphism were conducted by international European teams (Germany and Italy) do not use this term. The term "European American" which is the counterpart of the term "African American" must imperatively replace the term "Caucasian" in the text of this article for the part concerning autosomal STRs, even if it is not used by the authors of the US reference article. Citing a scientific reference is not an endorsement of its inappropriate terminology. In addition, the authors rightly emphasize the need to use genetic databases of well-defined reference populations to establish the BGA. Is this the case here for the autosomal STRs dataset?  

Author Response

Thank you

Stefania Morelli

Round 2

Reviewer 2 Report

The authors did revise the submitted manuscript however, some more corrections have to be done still:

Table 2 - It still contains incorrect information!!! Please review the continent assignment of the predicted haplogroups thoroughly

Table 2 - Nevgen prediction - were the hgs predicted using general level or subclades? please clarify and consider editing after re-running

Author Response

thanck you 
